# Evidence for Dosage Compensation in *Coccinia grandis*, a Plant with a Highly Heteromorphic XY System

**DOI:** 10.3390/genes11070787

**Published:** 2020-07-13

**Authors:** Cécile Fruchard, Hélène Badouin, David Latrasse, Ravi S. Devani, Aline Muyle, Bénédicte Rhoné, Susanne S. Renner, Anjan K. Banerjee, Abdelhafid Bendahmane, Gabriel A. B. Marais

**Affiliations:** 1Laboratoire de Biométrie et Biologie Evolutive (LBBE), UMR5558, Université Lyon 1, 69622 Villeurbanne, France; cecile.fruchard@gmail.com (C.F.); helene.badouin@univ-lyon1.fr (H.B.); benedicte.rhone@cirad.fr (B.R.); 2Institute of Plant Sciences Paris Saclay (IPS2), University of Paris Saclay, 91405 Orsay, France; david.latrasse@ips2.universite-paris-saclay.fr (D.L.); ravi.devani@students.iiserpune.ac.in (R.S.D.); abdel.bendahmane@ips2.universite-paris-saclay.fr (A.B.); 3Biology Division, Indian Institute of Science Education and Research (IISER), Pune 411008, Maharashtra, India; akb@iiserpune.ac.in; 4Department of Ecology and Evolutionary Biology, University of California Irvine, Irvine, CA 92697, USA; amuyle@uci.edu; 5Institut de Recherche pour le Développement (IRD), Université Montpellier, DIADE, F-34394 Montpellier, France; 6Systematic Botany and Mycology, University of Munich (LMU), Menzinger Str. 67, 80638 Munich, Germany; renner@lmu.de

**Keywords:** dioecy, sex chromosomes, Y degeneration, sex-biased genes, cucurbits

## Abstract

About 15,000 angiosperms are dioecious, but the mechanisms of sex determination in plants remain poorly understood. In particular, how Y chromosomes evolve and degenerate, and whether dosage compensation evolves as a response, are matters of debate. Here, we focus on *Coccinia grandis*, a dioecious cucurbit with the highest level of X/Y heteromorphy recorded so far. We identified sex-linked genes using RNA sequences from a cross and a model-based method termed SEX-DETector. Parents and F1 individuals were genotyped, and the transmission patterns of SNPs were then analyzed. In the >1300 sex-linked genes studied, maximum X-Y divergence was 0.13–0.17, and substantial Y degeneration is implied by an average Y/X expression ratio of 0.63 and an inferred gene loss on the Y of ~40%. We also found reduced Y gene expression being compensated by elevated expression of corresponding genes on the X and an excess of sex-biased genes on the sex chromosomes. Molecular evolution of sex-linked genes in *C. grandis* is thus comparable to that in *Silene latifolia*, another dioecious plant with a strongly heteromorphic XY system, and cucurbits are the fourth plant family in which dosage compensation is described, suggesting it might be common in plants.

## 1. Introduction

Some 5 or 6% of the angiosperms, depending on the assumed total species number, have male and female sporophytes, a sexual system termed dioecy [1]. Transitions from other sexual systems towards dioecy are estimated to have occurred between 871 and 5000 times independently [1]. Chromosomes and sex determination have been studied in few dioecious plants, and microscopically distinguishable (heteromorphic) sex chromosomes have been reported in about 50 species only [2]. An important question is whether sex chromosomes evolve similarly in plants and animals. For example, the evolution towards heteromorphy might be common between both lineages, with an autosomal origin of the sex chromosomes, gradual recombination suppression between X and Y chromosomes, and genetic degeneration of the Y chromosome [2,3,4,5]. Animal Y chromosomes tend to shrink over time and can become tiny as in the old heteromorphic systems of mammals and *Drosophila* [3,4,6]. In plants, the size of the Y chromosomes might evolve in a non-linear way, with Y chromosomes of intermediate age being larger than their X counterparts, but very old Y chromosomes being smaller [2]. Systems with large Y chromosomes have not been found in animals (with the single exception of the large neo-Y of *Drosophila miranda*, [7]) and might be plant-specific; the reason for this is unknown. *Silene latifolia* has one such system and has been intensively studied (e.g., [8,9,10,11,12,13,14,15,16,17,18]). Studying more plant systems, in particular, those with larger-than-X Y chromosomes, is necessary to get a more precise picture of the evolution of heteromorphic sex chromosomes in plants.

An important aspect of the evolution of animal sex chromosomes is dosage compensation, which has been reported in old heteromorphic systems [19]. The genetic degeneration of the Y chromosome causes a global decrease of Y gene expression through gene loss and gene silencing [6]. Without compensation, this phenomenon would result in a dosage imbalance in males because many Y genes have reduced expression, or are not expressed at all, compared to females where both X copies are fully expressed [19]. Three such compensation mechanisms have been described: the mammalian system, the *Drosophila* system, and the *C. elegans* system [20]. The mechanism in the fruit fly appears to be straightforward, with a chromosome-wide doubling of X expression in males, re-establishing the proper dosage [20]. The mammal and *C. elegans* mechanisms are less straightforward, with an apparent doubling of expression on the X chromosome in both sexes and then a mechanism to correct expression in females/hermaphrodites (X-inactivation in mammals, downregulation of both Xs in *C. elegans*, [20]). In both lineages, dosage compensation seems to affect certain dosage-sensitive genes only [19,20,21,22,23,24,25]). In birds, dosage compensation also is local, affecting only a few dosage-sensitive genes [26,27].

In plants, dosage compensation was first documented in *S. latifolia* [15] although its existence was initially disputed as different patterns were observed for X/Y gene pairs and X-hemizygous genes (lacking Y copies) in RNA-seq-based studies [13,15,28]. However, a study comparing sex-linked genes found in BAC sequences and in RNA-seq data suggested that X-hemizygous genes were probably under-represented in RNA-seq-based studies of *S. latifolia* [29]. Partial genome sequencing of *S. latifolia* confirmed this finding and provided a large set of X-hemizygous genes [16]. This study also confirmed that partial dosage compensation does exist in *S. latifolia* and that a fraction of both the X/Y gene pairs and X-hemizygous genes are compensated [16]. Dosage compensation in *S. latifolia* seems to be achieved through an upregulation of the maternal X chromosome in both sexes, which is reminiscent of the scenario envisioned by Ohno [30] for the evolution of dosage compensation in mammals [17,31]. The precise molecular mechanism remains unknown [32], but epigenetic studies suggest that the two female X chromosomes bear different epigenetic marks, implying different expression regulation [33,34]. Additionally, evidence for dosage compensation has been found in *Silene otites*, which has a ZW system younger than the *S. latifolia* XY one [35], in *Rumex rothschildianus*, which has an XY_1_Y_2_ system that is 8–10 Myr old [36], and in *Cannabis sativa*, which has an XY system that is 20–30 Myr old [37]. An important question is whether dosage compensation is a general feature of plant sex chromosomes.

Here, we focus on *Coccinia grandis*, a species in the Cucurbitaceae with a large Y chromosome [38,39,40,41]. *C. grandis* is a perennial, tropical liana that can produce fruits in the first year but can reach stem diameters of >8 cm and ages of at least 20 years. The genus comprises 25 species, all of them dioecious. It belongs to the tribe Benincaseae, where its sister genus (*Diplocyclos*) consists of four monoecious species [42]. The *C. grandis* Y chromosome comprises about 200 Mb, making it four times larger than the X chromosome, mainly due to the accumulation of transposable elements (TEs) and satellite repeats, resulting in a 10% difference in the size of male and female genomes [39,40,41]. Plastid and mitochondrial-like sequences also have accumulated on the *C. grandis* Y chromosome. The repetitive fraction of the male and female genomes of *C. grandis* is mainly composed of Ty1 copia and Ty3 gypsy LTR elements. Of these elements, five are found in much greater abundance on the Y than on other chromosomes [40]. Despite this heteromorphy, the species has been estimated to be only 3.1 Myr old based on a molecular-clock model applied to a phylogeny with all 25 species [42]. *C. grandis* males and females show no morphological differences except in their flowers, but between 2 and 8% of the genes appear to be differentially expressed between males and females [43,44]. The species is a promising system in which to study sex chromosome evolution because of its relatively small genome size (~1 Gb) and its phylogenetic proximity to *Cucumis* [45], which includes the fully sequenced *Cucumis sativus* and *Cucumis melo* genomes [46,47,48,49]. However, no sex-linked genes have been identified so far and no reference genome (with identified sex chromosomes) is currently available in *Coccinia*. Information about the extent of gene loss, the degradation of Y gene expression, the existence of dosage compensation, the genomic distribution of sex-biased genes is thus currently missing.

Here, we identify sex-linked genes in *C. grandis* using a model-based method that we developed and termed SEX-DETector [50] and which uses RNA-seq data to genotype the parents and F1 individuals from a cross. For each SNP, the transmission from parents to offspring of each allele is analyzed. Sex-linkage or autosomal segregation types have typical patterns that the SEX-DETector is able to differentiate even when there are genotyping errors. For example, a bi-allelic SNP in which one allele is transmitted exclusively from father to sons, while the other is transmitted from both parents to all progeny will be identified as an X/Y SNP (with the male-specific allele being the Y allele). The information of all SNPs in a gene is then combined into a probability for the gene to be sex-linked. RNA-seq-based segregation analysis is both relatively cheap and efficient, and has been applied in several plant systems in which sex-linked genes have been identified successfully, initially using empirical methods [13,14,15,51,52]. More recently, SEX-DETector has been applied in *Cannabis sativa*, *Mercurialis annua*, several *Silene* species, and *Vitis sylvestris* [17,35,37,50,53,54,55]. Based on the detected sex-linked genes, we aimed to estimate the age of the sex chromosomes and test for Y degeneration, dosage compensation, and sex-biased genes in *Coccinia grandis*.

## 2. Material and Methods

### 2.1. Plant Material

RNA sequencing data were obtained from the progeny and the parents of a cross between a male and a female individual of the dioecious plant *C. grandis*, both grown in the experimental fields of IISER Pune, India. Seeds from the cross were collected as soon as the fruit matured. The 24 seedlings raised from these seeds took four to seven months to begin flowering, which allowed sexing of the individuals. Flower buds at early developmental stages 3–4 (defined in [56]) were sampled from plants being grown in the experimental fields. RNAs were isolated from 5 males (sons) and 5 females (daughters) from the F1 generation as well as from their parents.

### 2.2. RNA Sequencing

The flower buds were sent to IPS2 Paris, France using RNA later ICE kits by Thermo Fisher. Total RNA was extracted from 12 flower bud samples using Agilent’s spin column purification method, mRNA was isolated with Oligo-dT Beads from NEB, and RNA-seq libraries were constructed with the Directional Kit from NEB. Sequencing was performed at IPS2 Paris, France, with Illumina NextSeq500 following a paired-end protocol of library preparation (fragment lengths 100–150 bp, 75 bp sequenced from each end). RNA samples were checked for quality, individually tagged, and sequenced (see Appendix A for library sizes). The sequence data are available in the ENA database under the study accession number PRJEB39318.

### 2.3. De Novo Transcriptome Assembly

A reference transcriptome was built for *C. grandis* using Trinity [57] on the combined libraries of the 12 individuals (the parents and their ten offspring). For each individual, 100% identical reads, assumed to be PCR duplicates, were filtered out using the ConDeTri v2.3 trimming software [58]. Reads were then filtered out for sequencing adapters and low quality using ea-utils FASTQ processing utilities v1.04.636 [59]. Cleaned reads from all male and female individuals were combined and assembled with Trinity version 2.4.0 with default settings [57] leading to 276,225 contigs. Poly-A tails were removed from contigs using PRINSEQ v0.20.4 [60] with parameters -trim_tail_left 5 -trim_tail_right 5. rRNA-like sequences were removed using riboPicker version 0.4.3 [61] with parameters—i 90 -c 50 -l 50 and the following databases: SILVA Large subunit reference database, SILVA Small subunit reference database, the GreenGenes database, and the Rfam database. To ensure that X and Y gametologs are assembled in consensus contigs (required for the SEX-DETector analysis, [50]), Trinity components were merged using Cap3 [62], with parameter -p 90 and custom perl scripts. Coding sequences were predicted using Trinity TransDecoder version 3.0.1 [57] and including Pfam domain searches as ORF retention criteria. This assembly included 128,904 ORFs. To avoid mapping X and Y reads on separate contigs of the same gene, we chose to work on the longest ORF predicted per Trinity isoforms, which resulted in a final set of 82,699 contigs (see Table 1). BUSCO v3.0.2 (Benchmarking Universal Single-Copy Orthologs) was used to assess the completeness of our transcriptome according to conserved gene content from the Plant Dataset [63]. Results are shown for full assembly with all ORFs and for the longest ORF per Trinity isoforms, hereafter referred to as our reference transcriptome (128,904 and 82,699 contigs respectively) in Appendix A.

### 2.4. Functional Annotation and Gene Ontology Enrichment Analysis

De novo annotation of our transcriptome was performed using Trinnotate v3.1.0 [57] and resulted in 59,319 annotations for 82,699 contigs (71.73%). Gene Ontology (GO) was assessed using GOSeq [64] version 1.30.0 on R version 3.4.3 (2017-11-30) to identify over or under-represented GO terms (*p*-value cutoff = 0.05).

### 2.5. Inferring Sex-Linked Contigs

The raw Illumina reads were mapped on the reference transcriptome using BWA mapping [65] version 0.7.15 with the following commands: bwa aln -n 5 and bwa sampe. Mapping statistics are shown in Appendix A. Mapped reads were kept with Samtools version 1.3.1 and individual genotypes were predicted with reads2snps version 2.0.64 with paralog detection [66,67], option -aeb which allows alleles to have different expression levels, and -par 0 to avoid removal of paralogous positions by the paraclean program, which tends to filter out X/Y SNPs.

SEX-DETector version 1.0 (3rd September 2017) was used to infer contig segregation types using a stochastic expectation maximization (SEM) algorithm [50]. The detected SNPs were filtered using Perl scripts to retrieve the autosomal or sex-linked SNPs, when their posterior probability to be either autosomal or sex-linked was higher than 0.8. A contig was then inferred as sex-linked if its global probability of being sex-linked was higher than the probability of it being autosomal and if it at least had one sex-linked SNP without genotyping error. Amongst sex-linked genes, X-linked contigs without a detectable homologous Y-linked copy are called X-hemizygous. Sex-linked contigs with no Y expression were considered as X-hemizygous, the rest as X/Y.

### 2.6. Correcting Mapping Bias

To avoid biases towards the reference allele in expression level estimates, a second mapping was done with GSNAP [68], a SNP-tolerant mapping software (see Appendix A). A SNP file of X/Y SNPs identified in the first run of SEX-DETector was produced with home-made perl scripts as described in [17]. Raw Illumina reads were mapped with GSNAP version 2017-11-15 and parameters -m 10 and -N 1. Only uniquely mapped and concordant paired reads were kept for expression analysis. SEX-DETector was run a second time on this new mapping, and the new inferences were used afterwards for all analyses. No significant difference in the number of sex-linked inferences was observed: 1196 X/Y and 168 X-hemizygous contigs were found as shown in Table 2.

### 2.7. Estimating the Age of Sex Chromosomes

The X and Y ORF sequences were produced by SEX-DETector using only X/Y segregating SNPs, and pairwise dS was estimated by the codeml program implemented in the PAML suite [69] version 4.8 (see Appendix A). The age of the sex chromosomes can be obtained from the X/Y gene pairs with the highest synonymous divergence (the first to stop recombining, [3,4]). To get age estimates in millions years, we used three Brassicaceae molecular clocks: 1.5 × 10^−8^ substitution/synonymous site/year, derived from an assumed divergence time of *Barbarea* and *Cardamine* of 6.0 Myr [70], 7.1 × 10^−9^ substitutions/site/generation based on spontaneous mutations in *Arabidopsis thaliana* [71], and 4 × 10^−9^ substitutions/synonymous site/year, derived from a phylogeny calibrated with six Brassicales fossils [72]. We obtained the age estimates as follows: age (in years) = dSmax/rate, using the molecular clock of [70] and [72], and age (in number of generations) = dSmax/2μ, using the molecular clock of [71] and assuming a *C. grandis* generation time of 1 to 5.5 years.

### 2.8. Estimating Gene Loss

X-hemizygous genes (X-linked genes without detectable Y copies) have been used to infer the extent of gene loss on Y chromosomes. This only gives a rough idea of gene loss as X-hemizygous genes inferred by SEX-DETector comprise both genes with deleted or silenced Y copies (true lost Y genes) and genes with Y copies that are expressed in some tissues but not in the one used for RNA-seq (false lost Y genes). In addition, X-hemizygous contigs are inferred by SEX-DETector from X polymorphism, as explained in [50], whereas X/Y contig inference relies on fixed mutations. X-hemizygous contigs can therefore only be detected in contigs with X polymorphism, resulting in their underestimation [14]. We corrected for this by using the number of X-hemizygous contigs (168) relative to X/Y contigs with X polymorphism (424) that were listed in the output of SEX-DETector. Premature stop codons were detected using a custom script on X and Y alleles.

### 2.9. Analysis of Expression Level Differences between X and Y Alleles

#### 2.9.1. Allelic Expression Measurement

Normalized allelic expression of sex-linked contigs was computed as in [17] from SEX-DETector output. Expression of X and Y alleles was computed using reads spanning diagnostic X/Y SNPs only and were normalized using the library size and the number of studied SNPs in the contig. Normalized expression levels were lower in males compared to females in autosomal contigs, as seen in Appendix A. This may be because a small subset of genes are very highly expressed in developing male organs, as observed in other plants (e.g., [73]), resulting in an apparent lower expression of housekeeping genes after normalization for total library size. We applied a correction coefficient to male expression using the ratio of median male autosomal expression over median female autosomal expression, to have a comparable median expression in males and females for autosomal contigs. The same ratio was then applied to sex-linked contigs. The results of this correction are shown in Appendix A. We used corrected expression levels to prepare Figure 1 and to perform all downstream analyses.

#### 2.9.2. Analysis of Dosage Compensation in X-Hemizygous Contigs

Contigs with Xmale/2Xfemale ratios above 8 or under 0.125 were excluded as in [15] because we do not expect dosage compensation to occur in these sex-biased genes. To test for dosage compensation, we filtered genes based on their log_2_ male-to-female expression ratio following [74]. X-hemizygous contigs that showed the same expression level in males and in females, i.e., with of log_2_(male/female) of 0 ± 0.2 were considered as compensated and X-hemizygous contigs with twice as much expression in females compared to males, i.e., with of log_2_(male/female) of −1 ± 0.2 were considered as non-compensated (see Figure 2). A wider range of log_2_(male/female) of ±0.5, as used in [27] for dosage-compensated contigs, had the same GO enrichment categories. A smaller range of ±0.1 did not have enough contigs per category to allow for a significant GO enrichment analysis.

### 2.10. Identifying Contigs with Sex-Biased Expression

We used the R packages DESeq2 [75] version 1.18.1, edgeR [76,77] version 3.20.8, and Limma-Voom [78] version 3.34.8 in R version 3.4.3 (2017-11-30) to perform biased gene expression analysis between males and females. DGE analysis was performed on the raw read counts (untransformed, not normalized for sequencing depth). Contigs with a CPM (count per million) lower than 0.5 (corresponding to a count of 10) were filtered out. Regularized log-transformation of the DESeq2 package was used to reduce variance of low read counts. Normalization with edgeR was made with a tagwise dispersion and GLM normalization method (calcNormFactors, estimateTagwiseDisp, and glmLRT functions). With LimmaVoom, counts were fitted to a linear model and differential expression was computed by empirical Bayes (lmFit and eBayes functions).

Contigs identified with at least two methods with an FDR cutoff of 0.0001 were retained as differentially expressed (see Appendix A). The three methods do not have the same characteristics, and keeping contigs identified with at least two methods as differentially expressed is more robust. DESeq2 and edgeR are based on the same method, but edgeR has fewer false positives. LimmaVoom is very specific (see Appendix A), and therefore also has few false positives. Keeping contigs identified with at least two methods is a way to both remove false negatives of DESeq2 and EdgeR and retain true positives that LimmaVoom tends to discard.

### 2.11. Statistics

Unless stated otherwise in the relevant sections, all statistical analyses and graphs were done with R version 3.4.3 [79]. Fisher’s exact tests were two-tailed and *p*-values were adjusted with the FDR method [80]. Exact adjusted *p*-values are provided for each test.

## 3. Results

### 3.1. Sex-Linked Genes Identified by SEX-DETector

We assembled a de novo transcriptome for *C. grandis* with male and female reads (82,699 contigs, see Table 1). We used RNA-seq data from a *C. grandis* F1 cross mapped to our reference transcriptome to identify genes located on the sex chromosomes (Appendix A). The raw reads were mapped on open reading frames (ORF). A total of 45.76% reads were mapped with standard mapping and 49.24% with SNP-tolerant mapping (see Appendix A). We divided the contigs expressed in buds into autosomal, sex-linked X/Y (defined as contigs having both X- and Y-linked alleles), and X-hemizygous contigs (sex-linked, but with no Y-copy expression). These categories were inferred from single nucleotide polymorphisms (SNPs) segregating in a family, using a probabilistic model [50]. Out of the 82,699 contigs, 5070 had enough informative SNPs to be assigned to a segregation type, 3706 were inferred as autosomal (73.10% of contigs with enough informative SNPs), 1196 as X/Y (23.59%), and 168 as X-hemizygous (3.31%) (see Table 2).

### 3.2. Age of the C. grandis XY System

Age estimates are based on the divergence between X and Y copies and three Brassicaceae molecular clocks (see Table 3). We obtained a maximum dS of 0.17 in all contigs and of 0.13 in contigs longer than 1kb. The molecular clocks obtained from fossil-calibrated Brassicaceae phylogenies [70,72] returned age estimates between 8.7 and 34.7 Myr old. The molecular clock obtained from *Arabidopsis thaliana* substitution rates [71] returned estimates ranging from 9.3 to 12.1 Myr, when assuming a generation time of 1 year. The true generation time of *C. grandis*, however, is unknown. One year is the onset of sexual maturity in this plant and corresponds to a lower bond estimate of the generation time. *Coccinia grandis* can reach 20 years in the wild, and individuals of 10 years are known from botanical gardens (S.S. Renner, pers. com). Assuming an average generation time of 5.5 years (from the minimum value of 1 year and the conservative maximum value of 10 years), yields much higher age estimates for XY divergence (see Table 3).

### 3.3. Patterns of Y Degeneration

We looked for patterns of degeneration in our data. Males showed lower gene expression than females for most of the genes, because a small subset of genes are very highly expressed in developing male flower buds, resulting in apparent lower mean expression after normalization for total library size (Appendix A). After correcting for this expression bias between males and females, we found that sex-linked genes are less expressed in males than in females (see Appendix A), Wilcoxon ranked test *p*-value = 2.57 × 10^−8^). To refine our analysis of Y chromosome degeneration in *C. grandis*, we analyzed the allelic expression of genes inferred as sex-linked, which showed that Y-linked alleles were significantly less expressed than X-linked alleles in males (see Figure 1, Wilcoxon ranked test *p*-value = 3.06 × 10^−13^). Lost Y genes can be detected by SEX-DETector when the Y copy is absent or unexpressed and are assigned as X-hemizygous. But given that X-hemizygous contigs can be inferred from segregation patterns only if there is polymorphism on the X chromosome [14,50], their number may be underestimated when segregation patterns are analyzed. We corrected for this as described in Materials and Methods, which resulted in a corrected rate of gene loss of 39.62%. Another hallmark of degeneration is the presence of premature stop codons. We detected 17 X (1.4%) and 56 Y (4.7%) alleles with a premature stop, implying that premature stop codons are more abundant in Y alleles (Fisher’s exact test *p*-value = 6.9 × 10^−6^). These observations clearly point to a significant level of genetic degeneration on the Y chromosome in *C. grandis*.

### 3.4. Patterns of Dosage Compensation

To determine whether some genes are dosage-compensated, we first studied the log_2_ fold change between male and female expressions. In the absence of dosage compensation, the Xmale/2Xfemale expression ratio is expected to be 0.5, so the log_2_ of the ratio is expected to be −1, because males (XY) have one X-linked copy and females (XX) have two. This is what we observed for contigs that do not show reduced expression of the Y-linked allele relative to the X-linked allele, i.e., that have a Y/X expression ratio close to 1 (median of log_2_ Xmale/2Xfemale ratio is −1.29 for contigs with Y/X > 1; see Figure 2). For contigs with reduced Y expression (low Y/X ratios), we observed a higher Xmale/2Xfemale expression ratio, which suggests that dosage compensation occurs for some genes (median of contigs with Y/X ≤ 0.5 is −0.85; see Figure 2). Finally, in X-hemizygous contigs, the distribution of Xmale/2Xfemale expression ratio was bimodal, with a set of 31 contigs centered on a log_2_ Xmale/2Xfemale ratio of −1 (no compensation), and 23 contigs centered on 0 (total compensation). This suggests that these two sets of genes exhibited respectively no compensation and total compensation resulting in equal expression in males and females. This trend is also present in a less visible pattern for X/Y contigs with low Y expression (see Figure 2). To investigate dosage compensation further, we compared expression of X-linked and Y-linked alleles in males and females for different Y/X expression ratio categories (Figure 3), using female expression as a reference. We excluded 1% of the sex-linked contigs that showed either an elevated Y expression (high Y/X ratios) or sex-biased X expression (very high or very low Xmale/2Xfemale ratios). The Y/X ratio was computed in *C. grandis* males and averaged between individuals, and used as a proxy for Y degeneration. In the absence of dosage compensation, Xmale/2Xfemale expression ratio is expected to be 0.5. Instead, we found that X expression in males increases with decreasing Y expression, which results in similar expression levels of sex-linked contigs in both sexes and provides further evidence for dosage compensation in *C. grandis*.

X/Y genes appeared to be depleted in hormone-related functions, such as response to ethylene and negative regulation of ethylene biosynthetic process (GO:0009723 and GO:0010366). X-hemizygous genes were depleted in dosage-sensitive functions such as macromolecular complex, intracellular ribonucleoprotein complex, ribonucleoprotein complex, transferase complex, and membrane protein complex (GO:0032991, GO:0030529, GO:1990904, GO:1990234, and GO:0098796). However, these same dosage-sensitive functions were enriched in X-hemizygous genes that show dosage compensation when compared to all sex-linked genes, which suggests compensation targets dosage-sensitive genes.

### 3.5. Genomic Distribution of Sex-Biased Genes

We detected 3453 sex-biased genes with edgeR, 4881 with DESeq, and 538 with LimmaVoom (Appendix A). To establish a robust set of sex-biased genes, we retained genes that were identified as sex-biased with at least two methods (3273 genes). Among these, 2682 (81.94%) were male-biased and 591 (18.06%) female-biased. Genes with sex-biased expression were significantly over-represented among sex-linked genes (see Appendix A, Fisher’s exact test *p*-value < 2.2 × 10^−16^), with 241 out of 1364 sex-linked genes being sex-biased (17.67% of sex-linked genes, with respectively 13.42% and 4.25% having male and female-biased expression), and 206 out of 3706 autosomal genes being sex-biased (5.56% of them, 4.45% with male and 1.11% with female-biased expression). Out of the sex-biased genes that were localized on the sex chromosomes, 228 (94.61%) had a X/Y segregation type (181 male-biased and 47 female-biased), and only 11 male-biased genes and 2 female-biased were X-hemizygous. X-hemizygous contigs were not enriched for differentially expressed genes when compared to autosomal contigs (see Appendix A, Fisher’s exact test *p*-value = 0.2302), which might be due to the small sample of X-hemizygous genes. Gene ontology analysis revealed several biological processes that are significantly over-represented among female-biased genes and under-represented among male-biased genes, or vice versa. GO categories related to pollen production were enriched in male-biased genes, such as pectin catabolic process, pollen wall assembly, sporopollenin biosynthetic process, pollen exine formation, pectin metabolic process, pollination, anther wall tapetum development, pollen sperm cell differentiation, anther development, rejection of self pollen, and regulation of pollen tube growth (GO:0045490, GO:0010208, GO:0080110, GO:0010584, GO:0045488, GO:0009856, GO:0048658, GO:0048235, GO:0048653, GO:0060320, and GO:0080092). Functions related to hormone signaling were enriched in female-biased genes, such as response to auxin, auxin-activated signaling pathway, regulation of ethylene-activated signaling pathway, brassinosteroid mediated signaling pathway, auxin polar transport, auxin mediated signaling pathway involved in phyllotactic patterning, ethylene receptor activity, ethylene binding, auxin transport, jasmonic acid and ethylene-dependent systemic resistance, and ethylene mediated signaling pathway (GO:0009733, GO:0009734, GO:0010104, GO:0009742, GO:0009926, GO:0060774, GO:0038199, GO:0051740, GO:0038199, GO:0051740, GO:0060918, and GO:0009871).

## 4. Discussion

### 4.1. Coccinia grandis XY are of Intermediate Age, Similarly to other Highly Heteromorphic Plant Systems

The divergence between X and Y copies in *C. grandis* reaches 0.13 to 0.17, and Brassicaceae-derived molecular clocks returned age estimates from 8.7 to 34.7 Myr for their divergence (and up to >50 Myr old assuming a generation time of 5.5 years, see Table 3). By contrast, a phylogeny that included 24 species of *Coccinia* and six outgroup taxa, calibrated with the divergence time of *Coccinia* and *Diplocyclos* of 15 ± 2.6 Myr, yielded a divergence time of *Coccinia grandis* from its sister species (whose sex chromosomes have not been studied) of 3.1 Myr [42]. In the other plant species with a huge Y chromosome, *S. latifolia*, sex chromosomes have been dated to 11.5 Myr old based on mutation rate measurements [81] or instead <1 to 10 Myr using molecular clocks ([9]: Brassicaceae and *Ipomoea* rates yield ~10 Myr for the sex chromosomes of *S. latifolia*; [82]: fossil-calibrated Caryophyllaceae phylogeny yields a divergence time of <1 – 3.75 Myr for *S. latifolia* and its sister clade). *Silene latifolia* has a generation time of 1 year, and it is thus tempting to think that *C. grandis* sex chromosomes should be as old or older than the ones of *S. latifolia*, unless mutation rate is a lot higher in *C. grandis*, which would be unexpected [81]. 

### 4.2. Coccinia grandis Y Chromosome Degeneration is Moderate, with an Unusually Reduced Y Expression

Our results suggest that not only is the Y chromosome of *C. grandis* accumulating repeats as shown previously, but it is also losing genes and becoming silent. Out of the 1364 sex-linked genes, 168 did not have a Y-linked homolog, and we estimated a total gene loss of about 40%, which is similar to what has been found in *S. latifolia* using a methodology similar to this study as well as other methods [16,17,29,50]. The Y/X expression ratio, however, was lower than what has been found in *S. latifolia* (0.63 vs. 0.77 respectively, [15]). The difference in size between the sex chromosomes is larger in *C. grandis* compared to *S. latifolia* [39], which might indicate that the TE load is larger in the former. TEs trigger host defense mechanisms, such as DNA methylation to silence them. This can affect genes close to TE insertions and reduce their expression level [83,84]. The possibly higher TE abundance on the *C. grandis* Y chromosome might trigger a higher level of gene silencing and thus explain its unusually low Y/X expression ratio. To test this idea further, the full sequences of the sex chromosomes in *C. grandis* and a DNA methylation study of TEs and neighboring genes would be needed.

### 4.3. Coccinia grandis Exhibit Sex Chromosome Dosage Compensation, a Phenomenon Observed in Several Plant Systems

When analyzing the expression of X and Y copies in both sexes, we found that the reduction of the Y copy was compensated by increasing expression of the X copy to maintain similar expression of the pair in both sexes (Figure 3). This suggests that dosage compensation has evolved in *C. grandis*. Again, a very similar pattern has been found in *S. latifolia* [15]. Compensation is probably partial, and not all the sex-linked genes are compensated (Figure 2), something also observed in *S. latifolia* [16]. We mentioned above that X-hemizygous genes are underrepresented in RNA-seq-based studies such as this one, and conclusions about dosage compensation are therefore more difficult to draw for this category of genes (discussed in [29]). Here, we found that the signal of dosage compensation was weak in X-hemizygous genes when taken together (Figure 3) as observed in *S. latifolia* with the same approach [15]. However, when looking at genes individually as in [16], we found evidence that some X-hemizygous genes are fully compensated (see bimodal distribution in Figure 2D). Strikingly, those genes are enriched in dosage-sensitive functions in agreement with findings in animals [21,22,23]; [19,20,24,25,27]. Evidence for dosage compensation has been found so far in Caryophyllaceae, Cannabaceae, and Polygonaceae [15,16,17,35,36,37]. Cucurbitaceae are thus the fourth plant family in which dosage compensation is documented.

This study did not allow us to identify a mechanism explaining the patterns of dosage compensation that we observed. Studies of animal and plant aneuploids have revealed that immediate adjustments of the genetic networks with expression levels changing at both sex-linked and autosomal genes are possible [85,86,87,88,89,90,91,92]. Other studies have revealed that several specific mechanisms of dosage compensation can evolve in parallel. In humans, for example, strong selection has maintained some essential dosage-sensitive genes on the Y, despite this chromosome being extremely degenerated [93]. For some other dosage-sensitive genes, the Y copy has been lost but this has been compensated by up-regulation of the X copy in both sexes, and inactivation of one of the X in females [21,22]. Finally, for a number of highly expressed tissue-specific genes that have lost their Y copy, dosage compensation took place through a process of duplication of the X copy and translocation of the duplicate to an autosome [94]. In plants, this remains to be investigated. In *S. latifolia*, the best-studied plant for dosage compensation, there are still debates about how dosage compensation is achieved [17,32]. In the present study, we assumed that autosomal expression was unchanged by Y degeneration, and used autosomal expression to further normalize male and female expression of sex-linked genes. Future studies should check this assumption, and could include an outgroup to study the evolution of both autosomal and sex-linked expression in *C. grandis* as in [17,53].

### 4.4. Coccinia grandis Sex Chromosomes are Enriched in Sex-Biased Genes

We found that 4% of the genes expressed in *C. grandis* floral buds are sex-biased (Appendix A), with a total of 3273 sex-biased genes identified (2682 male-biased and 591 female-biased), in agreement with prior studies on sex-biased genes in *C. grandis* [43,44]. Our results support enrichment for pollen production-related functions in male-biased genes as previously found in *C. grandis* [43,44] and in other plant systems (reviewed in [5]). Female-biased genes, on the other hand, were significantly enriched for hormone-signaling functions. These observations suggest that sex-biased expression may have evolved to support contrasting biological functions in *C. grandis* females and males. Male-biased genes are also significantly more numerous than female-biased genes, a pattern that is common in dioecious plants (e.g., [5,53,95,96] but see [97,98,99]). Lastly, sex-biased genes were found both on autosomes and on sex chromosomes but the latter were significantly enriched in such genes, again a common pattern in dioecious plants (for review, [5]; see also [97,98]). The most probable hypothesis is that sexually antagonistic selection favors sex linkage of sex-biased genes involved in sexual dimorphism, but further work to identify the footprints of sexually antagonistic selection (such as in [53]) will be needed to test this idea in *C. grandis*.

## Figures and Tables

**Figure 1 genes-11-00787-f001:**
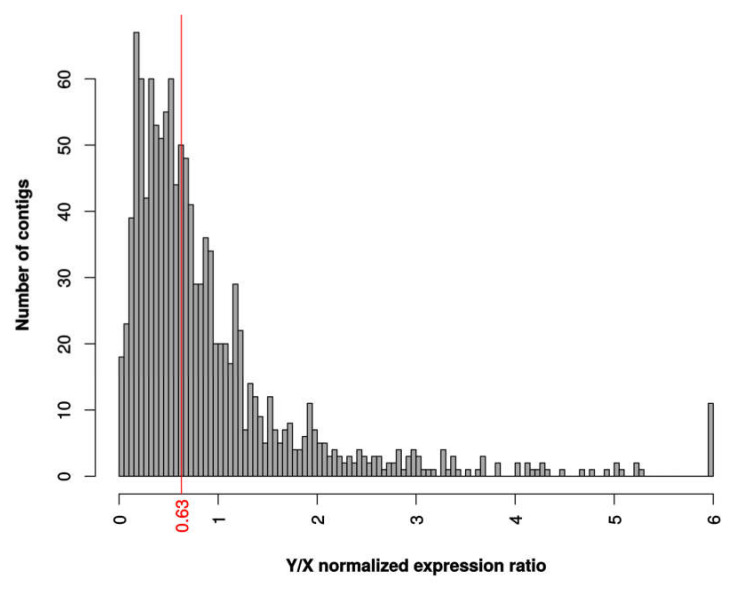
Y/X expression ratio in *C. grandis*. Distribution of normalized expression ratio between X and Y alleles. Total Y and X read numbers were summed at sex-linked SNP locations for each contig and normalized for each male separately, then averaged across males to obtain the Y/X ratio. The median is shown in red.

**Figure 2 genes-11-00787-f002:**
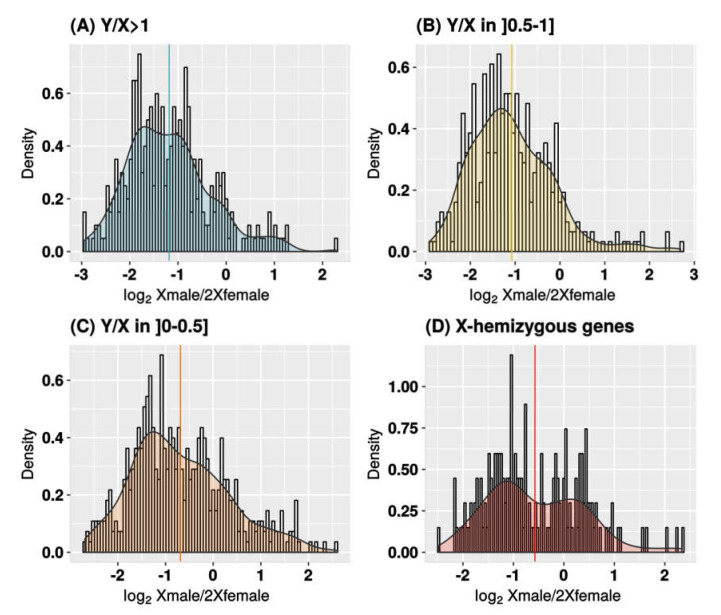
X expression in males versus females in *C. grandis*. Distribution of the ratio between the expression of the single X in males and the two X copies in females (log_2_ Xmale/2Xfemale) for all sex-linked contigs. Distributions are shown for Y/X expression ratio categories in males: (**A**) Y/X>1; (**B**) X/Y in ]0.5–1]; (**C**) X/Y in ]0–0.5]; (**D**) X-hemizygous genes (no Y copy expression). Total X read numbers were summed at sex-linked SNP locations in each contig and normalized for each individual separately, then averaged among males and females to get the Xmale/2Xfemale ratio. Distribution is shown in log_2_ scale with its density curve. Contigs with Xmale/2Xfemale ratios above 8 or under 0.125 were excluded, which reduced the dataset to 1351 sex-linked contigs. Sample sizes are: 0, 168; 0–0.5, 460 (8 outliers); 0.5–1, 389 (2 outliers); >1, 334 (3 outliers). Medians are indicated for each Y/X ratio category.

**Figure 3 genes-11-00787-f003:**
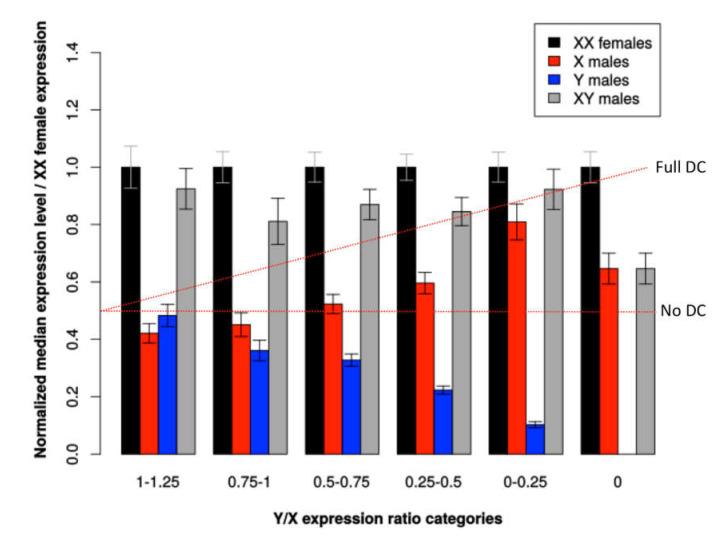
Allele-specific expression of sex-linked genes in both sexes in *C. grandis*. Expression levels of sex-linked contigs in both sexes are shown for different Y/X expression ratio categories. Total read numbers were summed at sex-linked SNP locations in each contig and normalized for each individual separately; medians for all contigs and individuals of the same sex were then obtained. XX females, median expression level of both X-linked alleles in females; X males, median expression level of the single X-linked allele in males; Y males, median expression level of the Y-linked allele in males; XY males, median expression level of the X-linked plus Y-linked alleles in males. To compare different Y/X expression ratio categories, medians were normalized using the XX expression levels in females. Two red lines indicate predictions for X males: full dosage compensation (X males = XX females − Y males) and no dosage compensation (X males = 0.5 XX females). With increasing Y degeneration (measured by the Y/X ratio), Y expression (blue bars) declines. We observed that X males (red bars) does not follow the prediction of no dosage compensation and instead tends to increase with increasing Y degeneration, following the ‘full dosage compensation’ line. The total expression in males (X males + Y males, grey bars) is mostly maintained across the different categories of Y degeneration and is comparable to the expression in females (black bars). These patterns suggest dosage compensation is taking place in *C. grandis*. The case of the X-hemizygous genes (Y/X = 0) is discussed in the text. Sample sizes are: 0, 168; 0–0.25, 207; 0.25–0.5, 261; 0.5–0.75, 243; 0.75–1, 148; 1–1.25, 108. Error bars indicate 95% confidence intervals of the median.

**Table 1 genes-11-00787-t001:** Transcriptome assembly statistics of *Coccinia grandis* flower buds. Statistics in the final Trinity transcriptome and the working transcriptome containing the longest ORF predicted per Trinity isoform.

	Full Transcriptome	Longest ORF per Isoform
Total contigs	128,904	82,699
Total assembled bases (bp)	103,275,123	27,290,670
Median contig length	552	836
Average contig length	801.18	836.83
Maximum contig length	16,296	16,296
Minimum contig length	297	297
N50	1029	1086
Total contigs longer than 1kb	30,795	21,587
GC content (%)	42.96	42.96

**Table 2 genes-11-00787-t002:** Results of the SEX-DETector pipeline on the *C. grandis* dataset. Number of contigs assigned by SEX-DETector to an autosomal, X/Y or X-hemizygous segregation type before and after SNP-tolerant mapping.

	SEX-DETector with BWA Mapping	SEX-DETector with GSNAP SNP-Tolerant Mapping
Contigs in final assembly	82,699	82,699
Contigs with enough coverage to be studied	82,689	70,298
Contigs with enough informative SNPs to compute a segregation probability	4320	3801
Contigs assigned to an autosomal segregation type	2889	3706
Contigs assigned to a X-Y segregation type	1239	1196
Contigs assigned to a X-hemizygous segregation type	192	168

**Table 3 genes-11-00787-t003:** Age estimates of the *C. grandis* XY system. These estimates were obtained using the maximum synonymous divergence between X and Y chromosomes and several molecular clocks from Brassicaceae. Estimates are shown in increasing order.

Molecular Clocks	Age Estimates of the Sex Chromosomes, with dS max = 0.17	Age Estimates of the Sex Chromosomes, with dS max = 0.13
From [70], calibrated with an assumed divergence time of *Barbarea* and *Cardamine* of 6.0 My	11.3	8.7
From [71], generation time = 1 year, assumed for *Arabidopsis thaliana*	12.1	9.3
From [71], generation time = 1.5 year	18.2	13.9
From [72], calibrated with six Brassicales fossils	34.7	26.5
From [71], generation time = 5.5 year	66.8	51.1

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
