# Peer review of "Evidence for Dosage Compensation in Coccinia grandis, a Plant with a Highly Heteromorphic XY System"

_genes, 2020, doi:10.3390/genes11070787_

Round 1

Reviewer 1 Report

The manuscript brings new information on dosage compensation between sexual chromosomes in Coccinia. The study is well constructed and all techniques and methods are sound.

The manuscript is well written, with just some details to improve:

all (see materials and methods) should be eliminated:

L27 correct Angiosperm, I would prefer the mechanisms

about instead of tilde

two keywords are already in the title

(refs 3,4,6, but see ref 7 for what?)

have reduced expression, or are not expressed at all (comma use)

see ref. 20). I guess eliminate see at all occurrences

X-hemizygous genes ??? in ref 16 it is loci. Please change at all occurrences.

This last study change in The last study

123 RNA sequencing data were obtained from a progeny of a cross

as soon as the fruit matured (no fruits)

leading to 276,225 contigs were obtained.

Y-linked copy are called X-hemizygous loci (I had rather to add loci). Sex-linked contigs with no Y expression were considered as X-hemizygous loci, the rest as X/Y loci.

Figure 2. change 1st bracket orientation

Last sentence of results section: Move to Discussion; These functional

enrichments suggest that sex-biased expression may have evolved to support contrasting biological functions in C. grandis females and males.

Author Response

see file

Reviewer 2 Report

This is an interesting MS, providing evidence for the dosage compensation in a highly heteromorphic XY flowering plant (Coccinia grandis) and analyzing its pattern.

I think it is a sound study bringing additional pieces to the huge plant sex-chromosome evolution puzzle. The findings from this study are obviously of interest to the sex-chromosome evolution community, especially plants, but also to the broader audience of this journal.

I would, though, have some general presentation and readability remarks:

The ms presents exhaustive and well-performed analyses, yet quite dense. So, to ease the reader's comprehension, I would advise reshaping the M&M section. To do so, the authors could lighten this section by moving some details and figures to supplementary data. It will also reduce some redundancy between the M&M and the Result section.

Fig 3 is a bit fuzzy.

A figure of hierarchical GO graphs comparing males and females triggered/enriched biological processes would be a plus.

Author Response

see file

Reviewer 3 Report

This paper documents the degeneration of the Y chromosome in the cucurbit, Coccinia grandis, and attempts to determine whether dosage compensation has occurred on the corresponding X chromosome. The paper is nicely written and the procedures seem reasonable. However, the authors make the assumption that autosomal expression would be unchanged with degeneration of the Y. Because the autosomes were used to normalize the total gene expression data, this is a potential flaw in the whole analysis (not possible to determine without further analysis). This assumption may or may not be true. Transcription factors and signal transduction components are now known to be dosage sensitive and to exert a quantitative modulation of their target genes across the genome. The most typical modulation in aneuploidy is an inverse effect (Birchler, 1979; 1981; Birchler and Newton, 1981; Birchler et al., 1990; Guo and Birchler, 1994; Sun et al., 2013a; b; Hou et al., 2018) (although other types of modulation certainly occur), so the normalization might obscure the X effects. The inverse effect could bring about dosage compensation (Birchler, 1981; Birchler et al., 1990; Hou et al., 2018). On the other hand, there could be retention of dosage sensitive regulatory genes (See Tasdighian et al 2017 for entry to gene dosage fractionation) on both the X and the Y as occurs in mammals and birds (Bellott et al 2014; 2017) resulting in the degeneration of the Y having minimal trans-acting effects on the X and autosomes. Yet, the results of the current analysis suggest that some genes on the X show a dosage effect and some are compensated. This is what occurs with aneuploidy of large regions in most species (Hou et al., 2018) so the conclusion could in fact be correct in general. However, if the autosomes were analyzed in the same way, such analysis might reveal that some that are up-regulated, some unaffected and some down-regulated (See Raznahan et al., 2018; Zhang et al., 2020). This would reveal whether the potential dosage compensation is similar to what happens in aneuploidy. Of course, it is possible that some other mechanism of dosage compensation has evolved. The authors take an X centric view, which is often the case by many authors, but is outdated (See Raznahan et al., 2018; Zhang et al., 2020). It would be good if the data were analyzed further to place the results in the context of dosage sensitive regulatory networks and the context of global effects of aneuploidy.

Birchler, J.A. (1979). A study of enzyme activities in a dosage series of the long arm of chromosome one in maize. Genetics 92: 1211–1229.

Birchler, J.A. (1981). The genetic basis of dosage compensation of alcohol dehydrogenase-1 in maize. Genetics 97: 625–637.

Birchler, J.A. and Newton, K.J. (1981). Modulation of protein levels in chromosomal dosage series of maize: the biochemical basis of aneuploid syndromes. Genetics 99: 247–266.

Birchler, J.A., Hiebert, J.C., and Paigen, K. (1990). Analysis of autosomal dosage compensation involving the alcohol dehydrogenase locus in Drosophila melanogaster. Genetics 124: 677–686.

Guo, M. and Birchler, J.A. (1994). Trans-acting dosage effects on the expression of model gene systems in maize aneuploids. Science 266: 1999–2002.

Sun, L., Johnson, A.F., Donohue, R.C., Li, J., Cheng, J., and Birchler, J.A. (2013a). Dosage compensation and inverse effects in triple X metafemales of Drosophila. Proc. Natl. Acad. Sci. U.S.A. 110: 7383–7388.

Sun, L., Johnson, A.F., Li, J., Lambdin, A.S., Cheng, J., and Birchler, J.A. (2013b). Differential effect of aneuploidy on the X chromosome and genes with sex-biased expression in Drosophila. Proc. Natl. Acad. Sci. U.S.A. 110: 16514–16519.

Raznahan, A. et al. (2018). Sex-chromosome dosage effects on gene expression in humans. Proc. Natl. Acad. Sci., U.S.A. 115: 7398-7403.

Tasdighian, S., Van Bel, M., Li, Z., Van de Peer, Y., Carretero-Paulet, L., and Maere, S. (2017). Reciprocally retained genes in the angiosperm lineage show the hallmarks of dosage balance sensitivity. Plant Cell 29: 2766–2785.

Hou, J. et al. (2018). Global impacts of chromosomal imbalance on gene expression in Arabidopsis and other taxa. Proc. Natl. Acad. Sci. U.S.A. 115(48): E11321-E11330..

Zhang, X. et al. (2020). Integrated functional genomic analyses of Klinefelter and Turner syndromes reveal global network effects of altered X chromosome dosage. Proc. Natl. Acad. Sci., U.S.A. 117: 4864-4873.

Bellott DW, et al. (2017) Avian W and mammalian Y chromosomes convergently retained dosage-sensitive regulators. Nat Genet 49:387-394.

Bellott DW, et al. (2014) Mammalian Y chromosomes retain widely expressed dosage-sensitive regulators. Nature 508:494-499.

Author Response

see file

Round 2

Reviewer 3 Report

The authors made some minor changes to the manuscript but might have made a stronger effort. A major point previously was that normalization to autosomal expression might give spurious results in terms of a conclusion of dosage compensation or not. Indeed, figures in the supplement indicate that autosomal expression is not equal between the sexes. The authors attribute this to high expression of male specific expression of a few genes. But examining the log2 distributions in supplemental figure 3, it is difficult to tell if that is the case and, indeed, it does not appear to be the case. Male specific expression would be a cluster of genes with very high log2 expression compared to females in Suppl. Figure 3, but this is not obvious. But even for the highest log2 male/female ratios, it is not possible to determine from the manuscript if these are highly expressed as suggested by the authors. However, if the authors examined the data, this could be determined, for example, by looking at the magnitude of expression of those genes with log2 male/female ratios above 2.5 versus other categories. Further, inspection suggests that the distribution is not randomly distributed around the mean, so it seems likely that a complex effect on the autosomes is occurring. A statistical test of the normality of the autosomal distribution would be good if it is to be used as a normalization, although see below.

The authors claim that an outgroup is needed to determine whether there is an autosomal change in expression but this is not the case. One could perform an exogenous spike-in (ERCC, for example) to total RNA before preparation for sequencing, either by polyA selection or rRNA depletion. Then, the X and autosomal expression can be independently normalized to the sequence reads of the exogenous spike-in. In retrospect, of course, this cannot be done. But dividing the autosomal expression into various levels of expression and then producing the log2 distributions might reveal any patterns that are occurring as one approach to dissecting the issue.

What could be done is the following. Because dosage compensation or dosage effect is a question of expression between log2 -1 and +1, the strong outliers with very high or low log2 male/female ratios could be removed from the analysis. Then the remaining genes on the X or the autosomes could be analyzed separately with normalization to the remaining RPKM. This would give an indication of whether there is a real effect occurring on the autosomes and whether it is valid to normalize the X to the autosomes (although that would not be necessary with this approach).

Having noted all of that, the fact that the Figure 2D distribution is multi-modal suggests that some genes on the X are indeed dosage compensated. Also, the authors did qualify their conclusions with an added paragraph in the Discussion. There are two points in that paragraph that are best deleted. The sentence “This suggests that changes in expression at autosomal genes to compensate for expression reduction at some Y-linked genes belonging to the same genetic network may have happened in C. grandis” should be removed because there is no reason to suggest this and it might not be the case. Also, as mentioned above, it is not necessary to have an outgroup to determine whether autosomal expression is different between the sexes.

Author Response

see file.
